# Topical Coherence in LDA-based Models through Induced Segmentation

## Abstract

This paper presents an `LDA`-based model that generates topically coherent segments within documents by jointly segmenting documents and assigning topics to their words. The coherence between topics is ensured through a copula, binding the topics associated to the words of a segment. In addition, this model relies on both document and segment specific topic distributions so as to capture fine grained differences in topic assignments. We show that the proposed model naturally encompasses other state-of-the-art `LDA`-based models designed for similar tasks. Furthermore, our experiments, conducted on six different publicly available datasets, show the effectiveness of our model in terms of perplexity, Normalized Pointwise Mutual Information, which captures the coherence between the generated topics, and the Micro F1 measure for text classification.

## 1 Introduction

Since the seminal works of Hofmann (1999) and Blei et al. (2003), there have been several developments in probabilistic topic models. Many extensions have indeed been proposed for different applications, including ad-hoc information retrieval (Wei and Croft, 2006), clustering search results (Zeng et al., 2004) and driving faceted browsing (Mimno and McCallum, 2007). In most of these studies, the initial exchangeability assumptions of `PLSA` and `LDA`, stipulating that words within a document are interdependent, has led to incoherent topic assignments within semantically meaningful text units, even though the importance of having topically coherent phrases is generally admitted (Griffiths et al., 2005). More recently, (Balikas et al., 2016b) has shown that binding topics, so as to obtain more coherent topic assignments, within such text segments as noun phrases improves the performance (*e.g.* in terms of perplexity) of `LDA`-based models. The question nevertheless remains as to which segmentation one should rely on.

Furthermore, text segments can refer to topics that are barely present in other parts of the document. For example, the segment "*the Kurdish regional capital*" in the sentence[1] "*A thousand protesters took to the main street in Erbil, the Kurdish regional capital, to condemn a new law requiring all public demonstrations to have government permits.*" refers to geography in a document that is mainly devoted to politics. Relying on a single topic distribution, as done in most previous studies including (Balikas et al., 2016b), may prevent one from capturing those segment specific topics.

In this paper, we propose a novel `LDA`-based model that automatically segments documents into topically coherent sequences of words. The coherence between topics is ensured through *copulas* (Elidan, 2013) that bind the topics associated to the words of a segment. In addition, this model relies on both document and segment specific topic distributions so as to capture fine grained differences in topic assignments. A simple switching mechanism is used to select the appropriate distribution (document or segment specific) for assigning a topic to a word. We show that this model naturally encompasses other state-of-the-art `LDA`-based models proposed to accomplish the same task, and that it outperforms these models over six publicly available collections in terms of perplexity, Normalized Pointwise Mutual Information (NPMI), a measure used to assess the coherence of topics with documents, and the Micro F1-measure in a text classification context.

---

[1] This sentence is taken from New York Times news (NYT) collection described in Section 4.

## 2 Related work

Probabilistic Latent Semantic Analysis (`PLSA`) proposed by (Hofmann, 1999) is the first probabilistic model that explains the generation of co-occurrence data using latent radom topics and, the EM algorithm for parameter estimation. The model was found more flexible and scalable than the Latent Semantic Analysis (Deerwester et al., 1990), which is based on the singular value decomposition of the document-term matrix, however `PLSA` is not a generative model as parameter estimation should be performed at each addition of new documents. To overcome this drawback, Blei et al. (2003) proposed the Latent Dirichlet Allocation (`LDA`) by assuming that the latent topics are random variables sampled from a Dirichlet distribution and that the generated words, occurring in a document, are exchangeable. The interdependence assumption allows the parameter estimation and the inference of the `LDA` model to be carried out efficiently, but it is not realistic in the sense that topics assigned to similar words of a text span are generally incoherent.

Different studies, presented in the following sections, attempted to remedy this problem and they can be grouped in two broad families depending on whether they make use of external knowledge-based tools or not in order to exhibit text structure for word-topic assignment.

### 2.1 Knowledge-based topic assignments

The main assumption behind these models are that text-spans such as sentences, phrases or segments are related in their content. Therefore, the integration of these dependent structures can help to discover coherent latent topics for words. Different attempts to combine `LDA`-based models with statistical tools to discover document structures have been successfully proposed, such as the study of Griffiths et al. (2005) who investigated the effect of combining a Hidden Markov Model with LDA to capture long and short distance dependencies. Similarly, (Boyd-Graber and Blei, 2008; Balikas et al., 2016a,b) integrated text structure exhibited by a parser or a chunker in their topic models. In this line, Du et al. (2013) following (Du et al., 2010) presented a hierarchical Bayesian model for unsupervised topic segmentation. This model integrates a boundary sampling method used in a Bayesian segmentation model introduced by Purver et al.(2006) to the topic model. For inference, a non-parametric Markov Chain inference is used that splits and merges the segments while a Pitman-Yor process (Teh, 2006) binds the topics. Recently, Tamura and Sumita (2016) extended this idea to the bilingual setting. They assume that documents consist of segments and the topic distribution of each segment is generated using a Pitman-Yor process (Teh, 2006).

Though, the topic assignments follow the structure of the text; these models suffer from the bias of statistical or linguistic tools they rely on. To overpass this limitation, other systems integrated automatically the extraction of text structure, in the form of phrases, in their process.

### 2.2 Knowledge-free topic assignments

This type of models extract text-spans using $n$-gram counts and word collections and use bigrams to integrate the order of words as well as to capture the topical content of a phrase (Lau et al., 2013). In (Wang et al., 2007), depending on the topic a particular bigram can be either considered as a single token or as two unigrams. Further, Wang et al. (2009) merged topic models with a unigram model over sentences that assigns topics to the sentences instead of the words.

Our proposed approach also does not make use of external statistical tools to find text segments. The main difference with the previous knowledge-free topic model approaches is that the proposed approach assigns topics to words based on two, segment-specific and document-specific distributions selected from a Bernoulli law. Topics within segments are then constrained using copulas that bind their distributions. In this way, segmentation is embedded in the model and it naturally comes along with the topic assignment.

## 3 Joint latent model for topics and segments

We define here a *segment* as a topically coherent sequence of contiguous words. By topically coherent, we mean that, even though words in a segment can be associated to different topics, these topics are usually related. This view is in line with the one expressed in (Balikas et al., 2016b), in which a latent topic model, referred to as `copLDA` in the remainder, includes a binding mechanism between topics within coherent text spans, defined in their study as noun phrases (NPs). The relation between topics is captured through a copula that provides

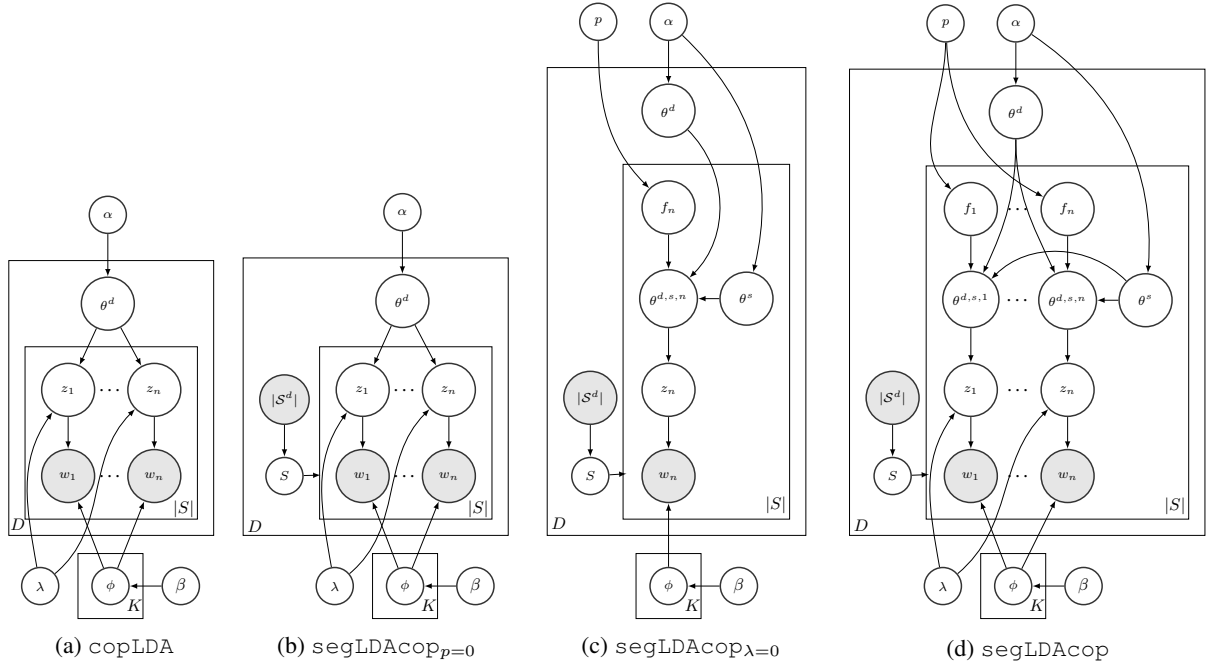

Figure 1: Graphical model for Copula LDA (`copLDA`), extension of Copula LDA with segmentation (`segLDAcop`$_{p=0}$), LDA with segmentation and topic shift (`segLDAcop`$_{\lambda=0}$) and complete model (`segLDAcop`).

a joint probability for all the topics used in a segment. That is, to generate words in a segment, one first jointly generates all the word specific topics $z$ via a copula, and then generates each word in the segment from its word specific topic and the word-topic distribution $\phi$. Figure 1(a) illustrates this.

Copulas are particularly useful when modeling dependencies between random variables, as the joint cumulative distribution function (CDF) $F_{X_1,\cdots,X_n}$ of any random vector $\mathbf{X} = (X_1,\cdots,X_n)$ can be written as a function of its marginals, according to Sklar's Theorem (Nelsen, 2006):

$$F_{X_1,\cdots,X_n}(x_1,\cdots,x_p) = C(F_{X_1}(x_1),\cdots,F_{X_n}(x_n))$$

where $C$ is a copula. For latent topic models, as discussed in (Amoualian et al., 2016), Frank's copula is particularly interesting as (a) it is invariant by permutations and associative, as are the words and topics $z$ in each segment due to the exchangeability assumption, and (b) it relies on a single parameter (denoted $\lambda$ here) that controls the strength of dependence between the variables and is thus easy to implement. In Frank's copula, when the parameter $\lambda$ approaches 0, the variables are independent of each other, whereas when $\lambda$ approaches $+\infty$, the variables take the same value. For further details on copulas, we refer the reader to (Nelsen, 2006).

One important problem, however, with `copLDA` is its reliance on a predefined segmentation. Although the information brought by the segmentation based on NPs helps to improve topic assignment, it may not be flexible enough to capture all the possible segments of a text. It is easy to correct this problem by considering all possible segmentations of a document and by choosing the most appropriate one at the same time that topics are assigned to words. This is illustrated in Figure 1(b), where a segmentation $S$ is chosen from the set $\mathcal{S}^d$ of possible segmentations for a document $d$, and where each segment in $S$ are generated in turn. We refer to the associated model as `segLDAcop`$_{p=0}$ for reasons that will become clear later.

Another point to be noted about `copLDA` (and `segLDAcop`$_{p=0}$) is that the topics used in each segment come from the same document specific topic distribution $\theta^d$. This entails that, in these models, one cannot differentiate the main topics of a document from potential segment specific topics that can explain some parts of it. Indeed, some text segments can refer to topics that are barely present in other parts of the document; relying on a single topic distribution may prevent one from capturing those segment specific topics.

It is possible to overcome this difficulty by generating a segment specific topic distribution as illustrated in Figure 1(c) (this model is referred to

as segLDAcop$_{\lambda=0}$, again for reasons that will become clear later). However, as some words in a segment can be associated to the general topics of a document, we introduce a mechanism to choose, for each word in a segment, a topic either from the segment specific topic distribution $\theta^s$ or from the document specific topic distribution $\theta^d$ (this mechanism is similar to the one used for routes and levels in (Paul and Girju, 2010)). The choice between them is based on the Bernoulli variable $f$, as explained in the generative story given below.

The above developments can be combined in a single, complete model, illustrated in Figure 1(d) and detailed below. We will simply refer to this model as segLDAcop.

### 3.1 Complete generative model

As in standard LDA based models, with $V$ denoting the size of the vocabulary of the collection and $K$ the number of latent topics, $\beta$ and $\phi^k$, $1 \leq k \leq K$, are $V$ dimensional vectors, $\alpha$ and $\theta$ (i.e., $\theta^d, \theta^s, \theta^{d,s,n}$) are $K$ dimensional vectors, whereas $z_n$ takes value in $\{1, \cdots, K\}$. Lower indices are used to denote coordinates of the above vectors. Lastly, $Dir$ denotes the Dirichlet distribution, $Cat$ the categorical distribution (which is a multinomial distribution with one draw) and we omit, as is usual, the generation of the length of the document. The complete model segLDAcop is then based on the following generative process:

1. Generate, for each topic $k, 1 \leq k \leq K$, a distribution over the words: $\phi^k \sim Dir(\beta)$;

2. For each document $d, 1 \leq d \leq D$:

   (a) Choose a document specific topic distribution: $\theta^d \sim Dir(\alpha)$;

   (b) Choose a segmentation $S$ of the document uniformly from the set of all possible segmentations $\mathcal{S}^d$: $P(S) = \frac{1}{|\mathcal{S}^d|}$;

   (c) For each segment $s$ in $S$:

      (i) Choose a segment specific topic distribution: $\theta^s \sim Dir(\alpha)$;

      (ii) For each position $n$ in $s$, choose $f_n \sim Ber(p)$ and set:

$$\theta^{d,s,n} = \begin{cases} \theta^s & \text{if } f_n = 1 \\ \theta^d & \text{otherwise} \end{cases}$$

      (iii) Choose topics $\mathcal{Z}^s = \{z_1, \ldots, z_n\}$ from Frank's copula with parameter $\lambda$ and marginals $Cat(\theta^{d,s,n})$;

   (iv) For each position $n$ in $s$, choose word $w_n$: $w_n \sim Cat(\phi^{z_n})$.

As on can note, the generative process relies on a segmentation uniformly chosen from the set of possible segmentations (step *2.b*) to generate related topics within each segment (Frank's copula in step *2.c.(iii)*), the distribution underlying each word specific topic $z_n$ being either specific to the segment or general to the document (steps *2.c.(i)* and *2.c.(ii)*). The other steps are similar to the standard LDA steps.

As in almost all previous studies on LDA, $\alpha$ and $\beta$ are considered fixed and symmetric, each coordinate of the vector being equal: $\alpha_1 = \cdots = \alpha_K$. The hyperparameters $p$ ($\in [0,1]$) of the Bernoulli distribution and $\lambda$ ($\in [0, +\infty]$) of Frank's copula respectively regulate the choice between the segment specific and the document specific topic distributions and the strength of the dependence between topics in a segment. As for the other hyperparameters, we consider them fixed here (the values for all hyperparameters are given in Section 4).

As mentioned before, all the models presented in Figure 1 are special cases of the complete model segLDAcop: hence segLDAcop$_{\lambda=0}$ is obtained by dropping the topic dependencies, which amounts to setting $\lambda$ to (a value close to) 0, segLDAcop$_{p=0}$ is obtained by relying only on the topic distribution obtained for the document, which amounts to setting $p$ to 0, and the previously introduced copLDA model is obtained by setting $p$ to 0, and fixing the segmentation.

### 3.2 Inference with Gibbs sampling

The parameters of the complete model can be directly estimated through Gibbs sampling. The Gibbs updates for the parameters $\phi$ and $\theta$ are the same as the ones for standard LDA (Blei et al., 2003). The parameters $f_n$ are directly estimated through: $f_n \sim Ber(p)$. Lastly, for the variables $z$, we follow the same strategy as the one described in (Balikas et al., 2016b) and based on (Amoualian et al., 2016), leading to:

$$P(\mathcal{Z}^s | \mathcal{Z}^{-s}, W, \Theta, \Phi, \lambda) = p(\mathcal{Z}^s | \Theta, \lambda) \prod_n \phi_{w_n}^{z_n}$$

where $W$ denotes the document collection, and $\Theta$ and $\Phi$ the sets of all $\theta$ and $\phi^k$, $1 \leq k \leq K$, vectors. $p(\mathcal{Z}^s | \Theta, \lambda)$ is obtained by Frank's copula with parameter $\lambda$ and marginals $Cat(\theta^{d,s,n})$. As is standard in topic models, the notation $-s$ means excluding the information from $s$.

From the above equation, one can formulate an acceptance/rejection algorithm based on the following steps: (a) sample $\mathcal{Z}^s$ from $p(\mathcal{Z}^s|\Theta, \lambda)$ using Frank's copula, and (b) accept the sample with probability $\prod_n \phi_{w_n}^{z_n}$, where $n$ runs over all the positions in segment $s$.

## 3.3 Efficient segmentation

As topics may change from one sentence to another, we assume here that segments cannot overlap sentence boundaries. The different segmentations of a document are thus based on its sentence segmentations. In the remainder, we use $L$ to denote the maximum length of a segment and $g(M; L)$ to denote the number of segmentations in a sentence of length $M$, each segment comprising at most $L$ words.

Generating all possible segmentations of a sentence and then selecting one at random is not an efficient process as the number of segments rapidly grows with the length of the sentence. In practice, however, one can define an efficient segmentation on the basis of the following proposition, the proof of which is given in Appendix A:

**Proposition 3.1.** *Let $l_i^s$ be the random variable associated to the length of the segment starting at position $i$ in a sentence of length $M$ (positions go from 1 to $M$ and $l_i^s$ takes value in $\{1, \cdots, L\}$). Then $P(l_i^s = l) := \frac{g(M+1-i-l;L)}{g(M+1-i;L)}$ defines a probability distribution over $l_i^s$.*

*Furthermore, the following process is equivalent to choosing sentence segmentations uniformly from the set of possible segmentations.*

```
From pos. 1, repeat till end of sentence:
(a) Generate segment length acc. to P;
(b) Add segment to current segmentation;
(c) Move to position after the segment.
```

In practice, we thus replace steps *2.b* and *2.c* of the generative story by a loop over all sentences, and in each sentence use the process described in Prop. 3.1. Furthermore, as described in Appendix A, the values of $g$ needed to compute $P(l_i^s = l)$ can be efficiently computed by recurrence.

## 4 Experiments

We conducted a number of experiments aimed at studying the impact of simultaneously segmenting and assigning topics to words within segments using the proposed segLDAcop model.

|  | Wiki0 | Wiki1 | Wiki2 |
|---|---|---|---|
| # words | 32,354 | 70,954 | 103,308 |
| *– vocabulary size* | 7,853 | 12,689 | 14,715 |
| # docs | 1,014 | 2,138 | 3,152 |
| *– maximal length* | 100 | 100 | 100 |
| # labels | 17 | 42 | 53 |
|  | Pubmed | Reuters | NYT |
| # words | 104,683 | 192,562 | 237,046 |
| *– vocabulary size* | 12,779 | 10,479 | 17,773 |
| # docs | 2,059 | 6,708 | 2,564 |
| *– maximal length* | 75 | 50 | 200 |
| # labels | 50 | 83 | - |

Table 1: Dataset statistics.

**Datasets:** We considered six publicly available datasets derived from Pubmed[2] (Tsatsaronis et al., 2015), Wikipedia (Partalas et al., 2015), Reuters[3] and New York Times (NYT)[4] (Yao et al., 2016). The first two collections were considered in (Balikas et al., 2016a), we followed their setup by considering 3 subsets of Wikipedia with different number of classes (namely, Wiki0, Wiki1 and Wiki2). The Reuters dataset comes from Reuters-21578, Distribution 1.0 as investigated in (Bird et al., 2009) and the NYT dataset is collected from full text of New York Times global news, from January 1st to December 31st, 2011.

These collections were processed following (Blei et al., 2003) by removing a standard list of 50 stop words, lemmatizing, lowercasing and keeping only words made of letters. To deal with relatively homogeneous collections, we also removed documents that are too long. The statistics of these datasets, as well as the admissible maximal length for documents, in terms of the number of words they contain, can be found in Table 1.

**Settings:** We compared our models (segLDAcop$_{p=0}$, segLDAcop$_{\lambda=0}$, segLDAcop) with three models, namely the standard LDA model, and two previously introduced models aiming at binding topics within segments:

1. LDA: Standard Latent Dirichlet Allocation implemented using collapsed Gibbs sampling inference (Griffiths and Steyvers, 2004)[5]. Note that there are neither segmentation nor topic

---

[2]https://github.com/balikasg/topicModelling/tree/master/data
[3]https://archive.ics.uci.edu/ml/datasets/Reuters-21578+Text+Categorization+Collection
[4]https://github.com/yao8839836/COT/tree/master/data
[5]http://gibbslda.sourceforge.net

| Models | Wiki0 | | Wiki1 | | Wiki2 | | Pubmed | | Reuters | | NYT | |
|---|---|---|---|---|---|---|---|---|---|---|---|---|
| | 20 | 100 | 20 | 100 | 20 | 100 | 20 | 100 | 20 | 100 | 20 | 100 |
| LDA | 853.7 | 370.9 | 1144.6 | 541.1 | 1225.2 | 570.6 | 1267.8 | 628.7 | 210.6 | 118.8 | 1600.1 | 1172.1 |
| senLDA | 958.4 | 420.5 | 1236.7 | 675.3 | 1253.1 | 625.2 | 1346.3 | 674.3 | 254.3 | 173.6 | 1735.9 | 1215.3 |
| copLDA | 753.1 | 264.3 | 954.1 | 411.5 | 1028.6 | 420.6 | 1031.5 | 483.2 | 206.3 | 101.3 | 1551.5 | 1063.2 |
| segLDAcop$_{p=0}$ | 670.2 | 235.4 | 904.2 | 382.4 | 975.7 | 409.2 | 985.5 | 459.3 | 194.2 | 96.7 | 1504.2 | 1033.2 |
| segLDAcop$_{\lambda=0}$ | 655.1 | 222.1 | 890.3 | 370.2 | 949.2 | 404.3 | 971.3 | 451.2 | 190.1 | 91.3 | 1474.6 | 1014.3 |
| segLDAcop | **621.2** | **213.5** | **861.2** | **358.6** | **934.7** | **394.4** | **960.4** | **442.1** | **182.1** | **87.5** | **1424.2** | **992.3** |

Table 2: Perplexity with respect to different number of topics (20 and 100).

binding mechanisms in this model;

2. senLDA: Sentence LDA, introduced in (Balikas et al., 2016a), which forces all words within a sentence to be assigned to the same topic. The segments considered thus correspond to sentences, and the binding between topics within segments is maximal as all word specific topics are equal;

3. copLDA: Copula LDA, introduced in (Balikas et al., 2016b) already discussed before, which relies on two types of segments, namely NPs (extracted with the nltk.chunk package (Bird et al., 2009)) and single words. In addition, a copula is also used to bind topics within NPs, from the document specific topic distribution.

Both senLDA and copLDA implementations, can be found in https://github.com/balikasg/topicModelling.

In all models $\alpha$ and $\beta$ play a symmetric role and are respectively fixed to $1/K$, following (Asuncion et al., 2009). For copula based models, $\lambda$ is set to 5, following (Balikas et al., 2016b). As already discussed, $p$ is set to 0 for segLDAcop$_{p=0}$; it is set to 0.5 for segLDAcop so as not to privilege *a priori* one topic distribution (document or segment specific) over the other. For sampling from Frank's copula, we relied on the R copula package (Hofert and Maechler, 2011) [6]. We chose $L$ (the maximum length of a segment) using line search for $L \in [2, 5]$ and used $L = 3$ in all our experiments. Finally, to illustrate the behaviors of the different models with different number of topics, we present here the results obtained with $K = 20$ and $K = 100$.

We now compare the different models along three main dimensions: perplexity, use of topic representations for classification and topic coherence.

### 4.1 Perplexity

We first randomly split here all the collections, using 75% of them for training, and 25% for testing.

In order to see how well the models fit the data and following (Blei et al., 2003), we first evaluated the methods in terms of perplexity defined as:

$$Perplexity = \exp\left(\frac{-\sum_{d\in D}\sum_{w\in d}\log\sum_{k=1}^{K}\theta_k^d\phi_w^k}{\sum_{d\in D}|d|}\right),$$

where $d$ is a test document from the test set $D$, and $|d|$ is the total number of words in $d$, and $K$ is the total number of topics. The lower the perplexity is, the better the model fits the test data. Table 2 shows perplexities of different methods for $K = 20$ and $K = 100$ topics.

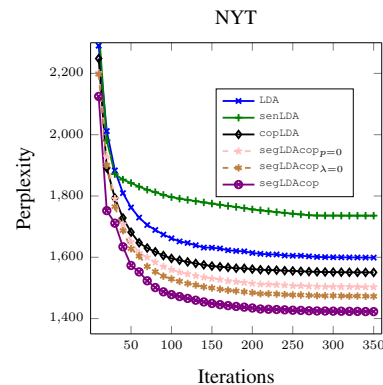

Figure 2: Perplexity with respect to training iteration on NYT collection (20 topics).

From Table 2, it comes out that the best performing model in terms of perplexity over all datasets and for different number of topics is segLDAcop. Further, segLDAcop$_{\lambda=0}$, that uses both document and segment specific topic distributions, performs better than segLDAcop$_{p=0}$, which in turn outperforms copLDA, bringing evidence that using all possible segmentations rather than only NPs unit extracted using a chunker yields a more flexible and natural topic assignment.

[6] Our complete code will be available for research purposes.

| Models | Wiki0 | | Wiki1 | | Wiki2 | | Pubmed | | Reuters | |
|---|---|---|---|---|---|---|---|---|---|---|
| | 20 | 100 | 20 | 100 | 20 | 100 | 20 | 100 | 20 | 100 |
| LDA | 55.3 | 63.5 | 42.4 | 51.4 | 41.2 | 48.7 | 54.1 | 63.5 | 75.5 | 82.7 |
| senLDA | 41.4 | 53.2 | 33.5 | 44.5 | 36.4 | 40.9 | 50.2 | 62.5 | 69.4 | 74.2 |
| copLDA | 51.2 | 62.7 | 43.4 | 52.1 | 40.8 | 46.5 | 53.5 | 63.1 | 75.2 | 81.5 |
| segLDAcop$_{p=0}$ | 59.1 | 64.2 | 44.8 | 51.2 | 42.3 | 50.1 | 55.4 | 63.1 | 76.8 | 82.5 |
| segLDAcop$_{\lambda=0}$ | 61.1 | 67.4 | 46.5 | 53.8 | 44.1 | 52.2 | 57.1 | 65.2 | 79.6 | 84.4 |
| segLDAcop | **62.3** | **68.4** | **48.4** | **55.2** | **44.8** | **53.5** | **59.3** | **66.5** | **80.2** | **85.1** |

Table 3: MiF score (percent) with respect to different number of topics (20 and 100).

segLDAcop also converges faster than the other methods to its minimum as it is shown in Figure 2, depicting the evolution of perplexity of different models over the number of iterations on the NYT collection (a similar behavior is observed on the other collections).

### 4.2 Topical induced representation for classification

Some studies compare topic models using extrinsic tasks such as document classification. In this case, it is possible to reduce the dimensionality of the representation space by using the induced topics (Blei et al., 2003). In this study, we first randomly splitted the datasets, except NYT that does not contain class information, into training (75%) and test (25%) sets. We then applied SVMs with a linear kernel; the value of the hyperparameter $C$ was found by cross-validation over the training set $\{0.01, 0.1, 1, 10, 100\}$. For datasets where certain documents have more than one label (Pubmed, Reuters), we used the one-versus-all approach for performing multi-label classification.

In Table 3, we report the Micro F1 (MiF) score of different models on the test sets. Again, the best results are obtained with segLDAcop, followed by segLDAcop$_{\lambda=0}$. This shows the importance of relying on both document and segment specific topic distributions. As conjectured before, our model is able to captures fine grained topic assignments within documents. In addition, all models relying on an inferred segmentation (segLDAcop$_{p=0}$, segLDAcop$_{\lambda=0}$, segLDAcop) outperform the models relying on fixed segmentations (sentences or NPs). This shows the importance of being able to discover flexible segmentations for assigning topics within documents.

### 4.3 Topic coherence

Another common way to evaluate topic models is by examining how coherent the produced topics are. Doing this manually is a time consuming process and cannot scale. To overcome this limitation the task of automatically evaluating the coherence of topics produced by topic models received a lot of attention (Mimno et al., 2011). It has been found that scoring the topics using co-occurrence measures, such as the pointwise mutual information (PMI) between the top-words of a topic, correlates well with human judgments (Newman et al., 2010). For this purpose an external, large corpus is used as a meta-document where the PMI scores of pairs of words are estimated using a sliding window. As discussed above, calculating the co-occurrence measures requires selecting the top-$N$ words of a topic and performing the manual or automatic evaluation. Hence, $N$ is a hyper-parameter to be chosen and its value can impact the results. Very recently, Lau and Baldwin (2016) showed that $N$ actually impacts the quality of the obtained results and, in particular, the correlation with human judgments. In their work, they found that aggregating the topic coherence scores over several topic cardinalities leads to a substantially more stable and robust evaluation.

Following the findings of Lau and Baldwin (2016) and using (Newman et al., 2010)'s equation, we present in Figure 3 the topic coherence scores as measured by the Normalized Pointwise Mutual Information (NPMI) . Their values are in [-1,1], where in the limit of -1 two words $w_1$ and $w_2$ never occur together, while in the limit of +1 they always occur together (complete co-occurrence). For the reported scores, we aggregate the topic coherence scores over three different topic cardinalities: $N \in \{5, 10, 15\}$. segLDAcop model which uses copulas and segmentation together, shows the best score for the given reference meta-data (Wikipedia) in all of the datasets. It should be noted that segLDAcop$_{\lambda=0}$ which has not copula binder inside the model has less improvement against the segLDAcop$_{p=0}$ which has the copula. This means using copula has more effect on the topic coherence than only the segment-specific topic distribution.

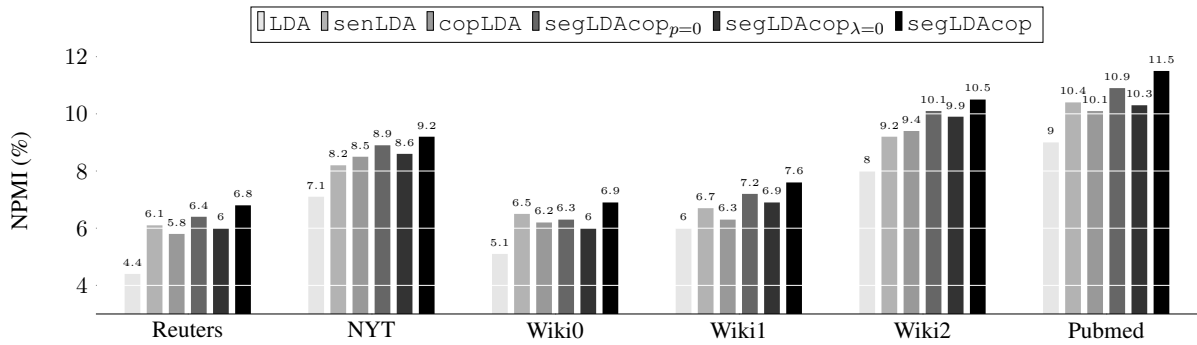

Figure 3: Topic coherence (NPMI) score with respect to 100 of topics.

## 4.4 Visualization

In order to illustrate the results obtained by `segLDAcop`, we display in Figure 4 the top 10 most probable words over 5 topics ($K = 20$) for the Reuters dataset, for both `segLDAcop` and `LDA`. In `segLDAcop`, topic 1, the top-ranked words are mostly relevant to the topic "date" (*e.g.*, march, january, year, fall, february, week). However, a similar topic learned by `LDA` appears to involve less such words (year, january, february), indicating a less coherent topic.

Figure 5 illustrates another aspect of our model, namely the possibility to detect topically coherent segments. In particular, as one can note, the NP *Rear Admiral Benjamin Sands* is segmented in two parts, the first one, *Rear Admiral*, corresponding to the title, and the second one, *Benjamin Sands*, to the first and family names of the person. The

| | | |
|---|---|---|
| **Topic1** | march, fell, rose, january, rise, year, fall, february, pct, week | fell, mln, year, january, dlrs, rise, rose, pct, billion, february |
| **Topic2** | currency, bank, pct, cut, rate, day, prime, exchange, interest, national | billion, prime, day, rate, dlrs, pct, reserve, federal, fed, bank |
| **Topic3** | term, agreement, acquire, buy, sell, unit, acquisition, corp, company, sale | term, dlrs, buy, company, sell, unit, corp, acquisition, sale, mln |
| **Topic4** | approved, american, common, split, merger, company, board, stock, share, shareholder | acquire, mln, company, common, stock, shareholder, share, corp, merger, dlrs |
| **Topic5** | tokyo, life, intent, letter, buy, insurance, yen, japan, dealer, dollar | central, european, japan, yen, ec, dollar, bank, rate, dealer, market |

Figure 4: Top-10 words of `segLDAcop` (left) vs `LDA` (right) for the Reuters (5 out of 20 topics).

Rear Admiral | Benjamin Sands | was an officer | in | the United States

Figure 5: Topic assignments with segmentation boundaries using `segLDAcop`. Colors are topics (examples from Wiki0 including stopwords with 20 topics).

data-driven approach we have adopted here can discover such fine grained differences, something the approaches based on fixed segmentations (either based on sentences or NPs), are less likely to achieve.

## 5 Conclusion

In this paper, we have introduced an `LDA`-based model that generates topically coherent segments within documents by jointly segmenting documents and assigning topics to their words. The coherence between topics is ensured through Frank's copula, that binds the topics associated to the words of a segment. In addition, this model relies on both document and segment specific topic distributions so as to capture fine grained differences in topic assignments. We have shown that this model naturally encompasses other state-of-the-art `LDA`-based models proposed to accomplish the same task, and that it outperforms these models over six publicly available collections in terms of perplexity, Normalized Pointwise Mutual Information (NPMI), a measure used to assess the coherence of topics with documents, and the Micro F1-measure in a text classification context. Our results confirm the importance of a flexible segmentation as well as a binding mechanism to produce topically coherent segments.

In the future, we plan on relying on other inference approaches, based for example on variational Bayes known to yield better estimates for perplexity (Asuncion et al., 2009); it is however not certain that the gain in perplexity one can expect from the use of variational Bayes approaches will necessarily result in a gain in, say, topic coherence. Indeed, the impact of the inference approach on the different usages of latent topic models for text collections remains to be better understood.

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

## A   Efficient segmentation

Let us recall the property presented before:

**Proposition A.1.** *Let $l_i^s$ be the random variable associated to the length of the segment starting at position $i$ in a sentence of length $M$ (positions go from 1 to $M$ and $l_i^s$ takes value in $\{1, \cdots, L\}$). Then $P(l_i^s = l) := \frac{g(M+1-i-l;L)}{g(M+1-i;L)}$ defines a probability distribution over $l_i^s$.*

*Furthermore, the following process is equivalent to choosing sentence segmentations uniformly from the set of possible segmentations.*

```
From pos. 1, repeat till end of sentence:
(a) Generate segment length acc. to P;
(b) Add segment to current segmentation;
(c) Move to position after the segment.
```

**Proof** Any segmentation of the sentence of length $M$ starts with either a segment of length 1, a segment of length 2, $\cdots$, or a segment of length $L$. Thus, $g(M; L)$ can be defined through the following recurrence relation:

$$g(M;L) = \sum_{l=1}^{L} g(M-l;L) \qquad (1)$$

together with the initial values $g(1; L), g(2; L), \cdots, g(L; L)$, which can be computed offline (for example, for $L = 3$, one has: $g(1; 3) = 1, g(2; 3) = 2, g(3; 3) = 4$). Note that $g(1; L) = 1$ for all $L$.

Thus:

$$\sum_{l=1}^{L} P(l_i^s = l) = \sum_{l=1}^{L} \frac{g(M + 1 - i - l); L)}{g(M + 1 - i; L)} = 1$$

due to the recurrence relation on $g$. This proves the first part of the proposition.

Using the process described above where segments are generated one after another according to $P$, for a segmentation $S$, comprising $|S|$ segments, let us denote by $l_1, l_2, \cdots, l_{|S|}$ the lengths of each segment and by $i_1, i_2, \cdots, i_{|S|}$ the starting positions of each segment (with $i_1 = 1$). One has, as segments are independent of each other:

$$P(S) = \prod_{j=1}^{|S|} P(l_{i_j}^s = l_j) = \prod_{j=1}^{|S|} \frac{g(M + 1 - (i_j + l_j); L)}{g(M + 1 - i_j; L)}$$

$$= \frac{g(M - l_1; L)}{g(M; L)} \frac{g(M - l_1 - l_2; L)}{g(M - l_1; L)} \cdots = \frac{1}{g(M; L)}$$

as $g(1; L) = 1$. This concludes the proof of the proposition. □

Furthermore, as one can note from Eq. 1, the various elements needed to compute $P(l_i^s = l)$ can be efficiently computed, the time complexity being equal to $O(M)$. In addition, as the number of different sentence lengths is limited, one can store the values of $g$ to reuse them during the segmentation phase.

