# Peer review of "Topical Coherence in LDA-based Models through Induced Segmentation"

_ACL 2017 — decision unknown_

[Official Review · Reviewer 1 · rating 4 · confidence 3]
soundness 3 · originality 3 · clarity 4 · impact 3 · substance 4 · appropriateness 4 · meaningful comparison 3 · presentation format Poster

- Strengths:
1. The idea of assigning variable-length document segments with dependent
topics is novel. This prior knowledge is worth incorporated in the LDA-based
framework.
2. Whereas we do not have full knowledge on recent LDA literature, we find the
part of related work quite convincing.
3. The method proposed for segment sampling with O(M) complexity is impressive.
It is crucial for efficient computation. 

- Weaknesses:
1. Compared to Balikas COLING16's work, the paper has a weaker visualization
(Fig 5), which makes us doubt about the actual segmenting and assigning results
of document. It could be more convincing to give a longer exemplar and make
color assignment consistent with topics listed in Figure 4.
2. Since the model is more flexible than that of Balikas COLING16, it may be
underfitting, could you please explain this more?

- General Discussion:
The paper is well written and structured. The intuition introduced in the
Abstract and again exemplified in the Introduction is quite convincing. The
experiments are of a full range, solid, and achieves better quantitative
results against previous works. If the visualization part is stronger, or
explained why less powerful visualization, it will be more confident. Another
concern is about computation efficiency, since the seminal LDA work proposed to
use Variational Inference which is faster during training compared to MCMC, we
wish to see the author’s future development.

[Official Review · Reviewer 2 · rating 4 · confidence 2]
soundness 3 · originality 3 · clarity 4 · impact 3 · substance 4 · appropriateness 5 · meaningful comparison 3 · presentation format Poster

### Strengths:
- Well-written, well-organized
- Incorporate topical segmentation to copula LDA to enable the joint learning
of segmentation and latent models
- Experimental setting is well-designed and show the superiority of the
proposed method from several different indicators and datasets

### Weaknesses:
- No comparison with "novel" segmentation methods

### General Discussion:
This paper presents segLDAcop, a joint latent model for topics and segments.
This model is based on the copula LDA and incorporates the topical segmentation
to the copula LDA. The authors conduct comprehensive experiments by using
several different datasets and evaluation metrics to show the superiority of
their model.

This paper is well-written and well-organized. The proposed model is a
reasonable extension of the copula LDA to enable the joint inference of
segmentations and topics. Experimental setting is carefully designed and the
superiority of the proposed model is fairly validated.
One concern is that the authors only use the simple NP segmentation and single
word segmentation as segments of the previous method. As noted in the paper,
there are many work to smartly generate segments before running LDA though it
is largely affected by the bias of statistical or linguistic tools used. The
comparison with more novel (state-of-the-art) segments would be preferable to
precisely show the validity of the proposed method.

### Minor comment
- In line 105, "latent radom topics" -> "latent random topics"